# Systemic Juvenile Idiopathic Arthritis/Pediatric Still’s Disease, a Syndrome but Several Clinical Forms: Recent Therapeutic Approaches

**DOI:** 10.3390/jcm11051357

**Published:** 2022-03-01

**Authors:** Pierre Quartier

**Affiliations:** 1Unité d’Immunologie-Hématologie et Rhumatologie Pédiatrique, RAISE Reference Centre, Hôpital Necker-Enfants Malades, Assistance Publique-Hôpitaux de Paris, 149 Rue de Sèvres, 75015 Paris, France; pierre.quartier@aphp.fr; 2INSERM 1231, Université de Paris, 45 Rue des Saints-Pères, 75006 Paris, France

**Keywords:** Systemic Juvenile Idiopathic Arthritis, Still’s disease, macrophage activation syndrome, interstitial lung disease, interleukin-1, interleukin-18, interleukin-6, biotherapy, Janus Kinase antagonists, allogeneic hematopoietic stem cell transplantation

## Abstract

Background: Systemic Juvenile Idiopathic Arthritis (SJIA)/Pediatric Still’s disease is associated with different phenotypes and outcomes from currently available treatments. Methods: A review of opinion, based on personal experience in a reference pediatric rheumatology center and key publications, to explore the most important questions regarding disease heterogeneity and treatment approaches. Results: A few situations deserve particular attention: 1/patients with recent-onset SJIA who may benefit from a treat-to-target approach with a key place for interleukin (IL)-1 inhibition; 2/SJIA patients refractory to Il-1 and IL-6 antagonists in whom several options may be discussed, including thalidomide or allogeneic hematopoietic stem cell transplantation; 3/SJIA patients with macrophage activation syndrome who may benefit from both well-used classical treatment and innovative approaches, such as anti-interferon gamma therapy or Janus Kinase (JAK) inhibitors; 4/SJIA with severe lung involvement, 5/SJIA patients who achieve complete remission on treatment, with some recent evidence that treatment may be reduced in intensity but not so easily withdrawn. Conclusions: a case-by-case discussion with expert teams is recommended in this heterogeneous, often difficult-to-treat population of patients.

## 1. Introduction

Systemic Juvenile Idiopathic Arthritis (SJIA) is defined according to the current International League of Associations for Rheumatology (ILAR) as a disease starting before the age of 16 years, active for at least 6 weeks, with at least 15 days of fever, a peculiar spiking fever pattern, arthritis, and at least 2 of the following features: skin rash, serositis (pericarditis in most cases), lymphadenopathy, hepatomegaly, or splenomegaly [1]. More recently, less stringent diagnosis criteria were proposed to allow earlier diagnosis and treatment [2]. These new diagnosis criteria are undergoing a validation process. A provisional diagnosis of SJIA could be made after two weeks in a patient with typical systemic features and arthralgia, even in the absence of arthritis.

The pathogenesis of SJIA is complex, with an autoinflammatory presentation involving proinflammatory cytokines such as interleukin (IL)-1 and Il-6, a risk of life-threatening macrophage activation syndrome (MAS) in some patients also involving other cytokines, such as interferon gamma and IL-18 [3,4,5,6,7,8,9,10]. In addition, there is a strong association of disease susceptibility and severity with the Human Leucocyte Antigen (HLA) DRB1 region [11,12,13,14]. There is a continuum with Adult-Onset Still’s Disease (AoSD), which diagnosis can be made in patients without arthritis [15,16]. In both SJIA and AoSD, some patients develop life-threatening complications such as MAS or secondary amyloidosis [17]. In addition, a subset of patients develops severe interstitial lung disease, alveolar proteinosis, and/or pulmonary hypertension [18,19].

Methotrexate and tumor necrosis factor (TNF) antagonists are not as effective as in non-systemic JIA [20,21]. Long-lasting corticosteroid therapy is associated with many complications, including growth failure in children. Therefore, there has been over the last year an increased usage of IL-1 and -6 antagonists, which have proven efficacious in controlling disease activity in most patients and allowing a substantial proportion of them to achieve inactive disease [22,23,24,25]. However, in the setting of clinical trials, stopping an effective biologic treatment in patients with clinical remission off steroids resulted in disease reactivation within a few months [26]. In some cases, the Il-1 receptor antagonist anakinra has been introduced in corticosteroid-naïve patients, after only a few weeks of disease duration, in an intention-to-treat approach [27]. More recently, phase III trials of the Janus Kinase (JAK) inhibitors, tofacitinib and baricitinib, have been initiated (NCT03000439 and NCT04088396).

SJIA is a syndrome rather than a disease, with some typical clinical features but very variable severity, outcomes, and variable responses to treatments. Peculiar gene expression signatures have been evidenced that distinguish SJIA from other diseases [3,22]. However, apart from a few cases, there is no underlying genetic abnormality. Importantly, recent efforts may allow the distinction of some subpopulations of SJIA patients who are at higher risk of severe complications and may deserve specific therapeutic approaches.

The author proposes to give his understanding of some key differences between different subpopulations of SJIA patients and their implications for patients’ prognosis and treatment.

## 2. Materials and Methods

This work is based on the author’s personal experience over 20 years in the fields of pediatric rheumatology, as the coordinator of a pediatric rheumatology team from a National reference center for rare pediatric inflammatory rheumatic diseases (RAISE, France) with a cohort of more than 200 SJIA patients, and from his knowledge of other experts’ experience, through regular case discussions, and of the literature.

## 3. Results

### 3.1. General Understanding of the Problematic Situations

Several publications emphasize that some situations in SJIA patients deserve peculiar attention [5,18,19,26,27]:-diagnosis and treatment of recent-onset SJIA;-treatment of patients with long-lasting auto-inflammatory symptoms who do not respond adequately to the most usual treatments;-treatment of patients with diffuse, severe, erosive polyarthritis;-treatment of patients with SJIA and macrophage activation syndrome (MAS), and in particular patients with remitting–relapsing MAS;-recognition and treatment of a subset of SJIA patients who may develop severe lung involvement, many of them having previously developed MAS;-tapering and/or withdrawing treatment in patients who achieve complete remission

Table 1 and Table 2 indicate some of the main questions (Table 1) and the classical versus more recent therapeutic approaches (Table 2) in several situations.

The differential diagnosis is particularly complex in the early phases of the disease, particularly with Kawasaki disease and post-Covid19 multisystem inflammatory syndrome [28].

### 3.2. Diagnosis and Treatment of Very Recent-Onset SJIA

Before the era of anti-IL-1 and anti-IL-6 targeted biologics, more than 50% of SJIA patients developed a chronic, severe disease, with persistent inflammation and erosive polyarthritis in most cases. On the other hand, between 15 and 40% of patients with a diagnosis of SJIA were reported to have a monocyclic disease course, or at least a complete remission after some weeks or months of active disease; the follow-up in these patients could last from three months to several years after remission [29,30,31,32].

Using the most recent diagnosis criteria, which allow a provisional diagnosis of SJIA to be made after two or three weeks of disease, even in the absence of arthritis [2], we may well select a subgroup of “SJIA” patients at lower risk of having a long-lasting, chronic course. A significant proportion of such patients may just have a post-infectious reactive disease that mimics SJIA at its onset but will vanish after a few weeks.

In the last few years, pediatric rheumatologists from Utrecht (NZ) and other places have hypothesized that there is a window of opportunity at the earliest phase of SJIA: at this stage, using an Il-1 inhibitor to treat steroid-naive patients would allow, in most cases, fast and complete remission, and prevent chronic inflammatory disease to develop [27]. Some authors suggested that such an approach should be considered within the first six weeks to three months of the disease [33,34,35]. Recent recommendations are in favor of a treat-to-target approach in JIA in general [36] and SJIA in particular [37], with very ambitious targets such as achieving control of fever and marked reduction of CRP level within one week, complete remission off steroids after some months; this has also stimulated a very active therapeutic approach of patients suspected of SJIA at an early phase of the disease, with often a very early introduction of the anti-IL-1 receptor antagonist anakinra.

The most recently published results from the team of Utrecht indicate that 12 out of 12 patients with no arthritis and 20 out of 30 with arthritis at anakinra onset achieved complete remission that persisted with the patient off treatment after 12 months and in most cases after 5 years [27]. Anakinra had been stopped in median after slightly more than three months.

Taking into account the absence of a control group and/or a randomization process, we do not know, in fact, if:-early anakinra treatment indeed modified the disease course in a significant proportion of patients, preventing diffuse, chronic polyarthritis and a long-lasting disease course;-or if some of these patients, particularly those who never developed arthritis, presented a less severe SJIA subtype (if not another disease) that would anyway have had a monocyclic course.

In the author’s, and many experts’, opinion, in patients with a provisional diagnosis of SJIA, early IL-1 blockade is an interesting approach. It certainly reduces the use of steroids [27,38]. It allows clinicians to meet more easily the most recent treat-to-target objectives [37]. It may also be valuable from a medico-economic point of view [39]. However, as with any major therapeutic decision in SJIA patients, a case-by-case discussion with an expert team is recommended. In particular, steroids remain important, possibly in association with anakinra, another biologic or a small molecule, in treating patients with an underlying SAM or early arthritis (Table 2). Patients with early arthritis may also benefit from early introduction of anti-IL-6 therapy, as anakinra may be less consistently effective in these cases [27]. Tocilizumab was also shown to be more efficient in such patients when started early and before the arthritis has spread to many joints [40]. Finally, although we lack reliable biomarkers to choose the best first line treatment, several polymorphisms in the *IL1RN* gene may be at risk of non-response to anakinra [35,41,42]. In addition, SJIA patients with very high IL-1 and IL-18 levels, particularly in cases of early-onset SJIA and SAM, might be good candidates for new therapeutic approaches, such as JAK inhibitors or other more experimental treatments targeting the interferon (IFN) gamma and/or IL-18 axis. In such patients, there is a high risk of non-response to treatments and life-threatening complications. Hence, they deserve a very tight follow-up, and even allogeneic hematopoietic stem cell transplantation (HSCT) may be worth considering in some cases, at a relatively early stage of the disease. This will be discussed in Section 3.6.

### 3.3. SJIA with Long-Lasting, Difficult-to-Treat, Inflammatory Disease

IL-1 and IL-6 antagonists have proven efficacy in most SJIA patients who would in the past have developed long-lasting inflammation and fail to adequately respond to methotrexate or other biologics such as TNF alpha antagonists. However, some patients either do not respond to, or escape symptomatic disease on anakinra, canakinumab or tocilizumab therapy. In patients who failed to respond to a TNF alpha antagonist, the response rate to anakinra seems good [43]; in patients who failed to achieve inactive disease on a first anti-IL-1 or Il-6 treatment, a significant proportion but not all patients eventually achieve inactive disease on a second or third-line biologic, either anti-IL-1 (mainly canakinumab in patients who did not respond well to anakinra or tocilizumab) or anti-IL-6 (tocilizumab in patients who did not respond well or escaped to anakinra and/or canakinumab) [25].

In patients with persistent auto-inflammation plus/minus arthritis, other options to discuss include JAK-inhibitors, thalidomide and, in a very few selected cases, allogeneic HSCT (Table 2) [44,45,46].

### 3.4. SJIA with Refractory Polyarthritis

SJIA patients with diffuse polyarthritis are among the most difficult to treat. Among currently available treatments, the best evidence-based medicine is for tocilizumab [24]. However, for the patients who do not respond well to IL-1 and Il-6 antagonists, the probability to achieve inactive or nearly inactive disease on any other drug and no or low-dose steroids is low. In this situation, it might be worth to test other biologics such as TNF inhibitors or abatacept in combination with methotrexate and, if needed, low-dose steroids. In some patients we used a combination of thalidomide and a TNF inhibitor (unpublished) with some good responders but only in a very few patients. The experience with JAK inhibitors in such patients is very recent. Again, in few selected cases, allogeneic HSCT has to be discussed (Table 2).

### 3.5. SJIA and Macrophage Activation Syndrome (MAS)

In patients with SJIA features and MAS at disease onset, several diseases have to be ruled out (Table 1). Even though MAS requires to be treated urgently in most case, having a few milliliters of blood sampled on EDTA at room temperature before starting high-dose steroids is important to allow proper lymphocyte phenotyping and analysis of the proportion of activated T cells; a high percentage of highly activated T cells is not in favor of SJIA-associated MAS but rather of primary hemophagocytosis such as familial lymphohistiocytosis or other rare conditions. Other analyses would then be performed such as studying the expression of perforin.

High-dose steroids remain the first line treatment in such patients to control MAS, however, adding other immuno-modulatory treatments has to be considered in most cases. Cyclosporine and, in life-threatening MAS, etoposide are the most classical treatments [47,48,49]. In patients who develop MAS after inappropriate discontinuation of an active SJIA treatment, such as tocilizumab or anakinra, reintroducing the active drug together with high-dose steroids must be considered. Of note, the diagnosis of MAS may be more challenging on anti-IL-6 treatment [50]. Some authors even propose the use of high-dose anakinra, in some cases intravenously, to treat MAS in the context of active SJIA activity [51]. On the other hand, several works are in favor of a major implication of IFNγ in SJIA-associated MAS [5,6,7]. The anti-IFNγ antibody, emapalumab, is being tested in this indication following its development in primary hemophagocytosis [52]; emapalumab seems active on MAS, however controlling associated SJIA activity requires additional therapy in most cases, including anakinra [53]. Targeting Il-18 may also be of interest in some patients [9,10]. There is an ongoing trial with a biclonal anti-IL-1/IL-18 antibody in patients with an *NLRC4* gain of function mutation, a rare autoinflammatory disease; such an approach might be also of interest in SJIA patients with MAS.

JAK inhibitors look promising as these drugs may be active on both SJIA activity and MAS [54,55,56]. In particular, in patients with remitting–relapsing MAS, we would at the moment privilege this treatment.

As for patients with primary hemophagocytosis, intractable MAS may lead to the proposal of allogeneic HSCT in a few SJIA patients [46].

### 3.6. SJIA at Risk of Life-Threatening Chronic Lung Disease

A subset of SJIA patients develop chronic, life-threatening lung disease including interstitial lung disease, pulmonary hypertension and alveolar proteinosis [18,19]. No underlying genetic abnormality has been found in these patients, however most of them have early-onset SJIA, permanently high serum IL-1 and IL-18 levels, a poor response to biologics and more hypersensitivity reactions to tocilizumab. The authors of the main publications suspect that macrophage dysfunction may be part of this severe outcome. We do not know if a less systematic usage of steroids at disease onset is responsible for a seemingly increased incidence of such severe lung involvement; the possible implication of environmental factors is also not clear. IFNγ has been reported to be essential for alveolar macrophage-driven pulmonary inflammation in macrophage activation syndrome [5,7]. A few case reports suggest that JAK inhibitors might be of interest in such patients [56].

### 3.7. Tapering and/or Withdrawing Treatment in SJIA Patients Who Achieve Complete Remission

In the Utrecht experience, most patients who received anakinra treatment at a very early stage of the disease were able to stop anakinra after 3.5 months in median and remained in remission, off steroids with a median follow-up of more than 5 years [27]. However, as indicated in Section 3.2, in the absence of any control group and in the absence of arthritis in a significant proportion of these patients with a “provisional diagnosis” of SJIA, it is not possible to know if there is a window of opportunity to avoid a chronic disease to develop, or if the patients with a favorable outcome had a reactive systemic disease instead of true SJIA or were just prone to have a monocyclic course.

In patients who achieved complete remission on the anti-IL-1 antibody canakinumab, which had been started two years in median after SJIA onset, a randomized controlled trial showed that most patients could reduce treatment intensity either by reducing the dosage from 4 to 2 and then 1 mg/kg per injection or by giving less frequent injections, up to 1 injection every 12 weeks; however, following canakinumab withdrawal, most patients experienced disease reactivation within 6 months, suggesting that a certain level of Il-1 inhibition might be required in such patients in the long term [26].

The experience with other drugs is much more limited and we clearly miss biomarkers to better predict which patients could stop treatment once they achieve complete remission, and which patients are at risk of flare-ups of disease.

## 4. Conclusions

SJIA remains a very complex, heterogeneous, difficult-to-treat condition in many patients, particularly those with long-lasting systemic disease and/or erosive polyarthritis, who do not respond well to Il-1 and Il-6 antagonists, and those with remitting–relapsing MAS. The risk of severe lung involvement in a subset of patients deserve particular attention. A case-by-case discussion with expert teams is recommended to properly diagnose and treat these patients, with some promising recent treatment such as JAK inhibitors or, in selected cases, of allogeneic HSCT, that deserve further investigation.

## Figures and Tables

**Table 1 jcm-11-01357-t001:** Peculiar situations.

Clinical Situations	Main Questions
**Recent-onset SJIA symptoms**	**Differential diagnosis:**
**Macrophage activation** **Syndrome (MAS)**	- Infections - Post-infectious (Covid19, …) - Vasculitis (Kawasaki, PAN, …) - Inflammatory bowel disease - Autoimmune disease (lupus, hepatitis, …) - Monogenic autoinflammatory syndrome - Neoplasia (rare) **Therapeutic emergency ^1^:** - Macrophage activation syndrome - Dyspnea on arythenoid arthritis (rare) - Myocarditis, marked pericarditis - Marked inflammation and pain - Early, diffuse polyarthritis **Differential diagnosis:** - Infection (EBV, Leishmania …) - ±monogenic immune disease (familial lymphohistiocytosis, Purtilo’s syndrome … or combined immunodeficiency) ^1^ - Neoplasia (T-, NK-cell lymphoma, …) **SJIA subtype at risk of lung disease:** - Early-onset SJIA, recurrent MAS - ±chromosomal abnormality (trisomy 21 …) - “hypersensitivity” to biologics **Urgent, ±intensive treatment ^1^:** - See Table 2

SJIA, systemic juvenile idiopathic arthritis; PAN, periarteritis nodosa; Epstein–Barr virus. ^1^ case by case discussion with an expert center recommended.

**Table 2 jcm-11-01357-t002:** Therapeutic approaches ^1^.

Clinical Situations	Classical Approach	Recent Approaches
**Recent-onset SJIA with auto-inflammatory syndrome**	±NSAIDs first	Anti-IL-1 treatment ^2^
	High-dose steroids	or anti-IL-6 treatment ±lower-dose steroids
**Long-lasting systemic inflammation with limited joint involvement**	Anti-IL-1 treatment Anti-IL-6 treatment	JAK-inhibitor
	±NSAID or low-dose steroids	In refractory cases, discuss: - thalidomide - allogeneic hematopoietic stem cell transplantation
**Severe MAS flare in a SJIA patient**	High-dose steroids ±cyclosporin	High-dose steroids associated with: - high-dose anakinra (±IV)
	±etoposide	- or anti-IFNγ (emapalumab) ± anakinra or another SJIA treatment ^3^ - or a JAK-inhibitor
**Remitting-relapsing MAS**	Steroids ± cyclosporine	JAK-inhibitor ^4^, steroids. In refractory cases, discuss: - anti-INFγ (emapalumab) - allogeneic hematopoietic stem cell transplantation
**Diffuse polyarthritis**	Anti-TFN treatment Anti-IL-1 treatment Anti-IL-6 treatment	Anti-IL-6 treatment (more evidence- based medicine than for other biologics or JAK-inhibitors)
	±methotrexate ± low-dose steroids	±methotrexate ±low-dose steroids In refractory cases, discuss allogeneic hematopoietic stem cell transplantation

SJIA, systemic juvenile idiopathic arthritis; NSAID, non-steroidal anti-inflammatory drug; IL, interleukin; JAK, janus kinase; MAS, macrophage activation syndrome; IFN, interferon. ^1^ case by case discussion with an expert center recommended. ^2^ more evidence-based medicine for anti-IL-1 treatment and in particular anakinra in this situation. ^3^ anti-IFNγ treatment may help controlling MAS but not the underlying systemic disease that may need other therapy in association. ^4^ may be active both on MAS and on the underlying systemic disease.

## Data Availability

Not applicable as this viewpoint is not a study reporting data.

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
