# Peer review of "Systemic Juvenile Idiopathic Arthritis/Pediatric Still’s Disease, a Syndrome but Several Clinical Forms: Recent Therapeutic Approaches"

_jcm, 2022, doi:10.3390/jcm11051357_

Round 1

Reviewer 1 Report

This review article is well-written informative paper. I have a few minor concerns as the following.

  1. Some comments regarding recent advances of pathogeneisis of systemic onset juvenile idiopathic arthritis should be added.
  2. Some description regarding proinflammatory cytokines in the pathogenesis and differential diagnosis should be discussed.

Author Response

Thank your for your comments.
I have added some comments and references regarding recent advances of pathogenesis of SJIA, proinflammatory cytokines (Second paragraph of the introduction)

I have added a sentence regarding the complexity of the differential diagnosis just before Table 1, that provides more information to this regards, with an additional reference (references N°28).

Reviewer 2 Report

In the current manuscript, the author discussed the different phenotypes and the response of the available treatments for Systemic Juvenile Idiopathic Arthritis. Author focus on some of the specific phenotypes/conditions of the SJIA which need specific attention. The author mentioned some of the classical approaches for SJIA treatment and also suggested some recently adopted treatment approaches based on own expertise and available literature support. Though, the manuscript focus on a complicated disease it is more of a review or opinion than an original article. The introduction and results are adequate and referred the relevant literature. The language of the manuscript is complicated at many places and needs improvement for better readability.

Author Response

Thank your for your comments.

This article is indeed rather a review of opinion (I was requested to prepare for a special issue of the Journal) than an original article, I have tried to make it more understandable in the Abstract and in the Material and Methods section.
I am sorry that the language of the manuscript is complicated and needs improvement. I have tried and shortened some sentences.

Reviewer 3 Report

1-The narrative review could be designed in a more intense methodology. There could be a specific search strategy. Different databases could be searched through. Inclusion and exclusion criteria regarding the literature should have been given.

2-The review could be supported by relevant figure presentations. 

3-The novelty of the paper is not clear. 

Author Response

Thank your for your comments.

This article is indeed a review of opinion (I was requested to prepare for a special issue of the Journal), not a meta analysis or a full review of the literature. I have tried to make it more understandable in the Material and Methods section.
I have not added figures as I feel the tables properly highlight the most important points.

This paper has no ambition of novelty for true experts of the disease, rather to bring some lights on its diversity and complexity for less expert colleagues.

Reviewer 4 Report

This review in interesting but a major revision of the entire manuscript is needed.

Introduction: please provide also epidemiological data. The introduction do not specify the aims of the review.

Material and Methods : please indicate the exact time of the period of the research from ... to... and indicate which article were excluded and why.

Results: how many article you checked ?

I suggest to construct better each paragraph to be more readable : first describe the first approach according to EULAR guideline, then describe compliance and possible therapies, and last personal experience 

Author Response

Thank your for your comments.

This article is a review of opinion I was requested to prepare for a special issue of the Journal, not a full review of the disease, its epidemiology, etc… I have tried to make it more understandable in the Introduction (last paragraph, that used to be the first paragraph of the Material and Methods section).
It is not a meta analysis or a full review of the literature. I have tried to make it more understandable in the Material and Methods section. Hence, I do not indicate in the results how many articles I checked.

Regarding your last suggestion, I do not feel the EULAR or the ACR guidelines would be a good start as the main purpose of this paper is not to issue recommendations, rather do bring some lights on its diversity and complexity for less expert colleagues.

Round 2

Reviewer 4 Report

According that is a viewpoint The article can be accepted